

# Constructing a neural network model based on tumor-infiltrating lymphocytes (TILs) to predict the survival of hepatocellular carcinoma patients

Wenqing Zhong[1], Ziyin Zhao[2], Xin Fang[3], Jingyi Sun[1], Yanbing Wei[1], Fengda Li[4], Bing Han[1] and Cheng Jin[3]

[1] Department of Hepatobiliary and Pancreatic Surgery, The Affiliated Hospital of Qingdao University, Qingdao, Shandong, China
[2] Organ Transplantation Center, The Affiliated Hospital of Qingdao University, Qingdao, Shandong, China
[3] School of Biomedical Engineering, Shanghai Jiao Tong University, Shanghai, Shanghai, China
[4] Department of Hepatobiliary Surgery, Gao mi People's Hospital, Weifang, Shandong, China

Corresponding authors
Bing Han, hanbing@qduhospital.cn
Cheng Jin, chengjin520@sjtu.edu.cn

## ABSTRACT

**Background:** Hepatocellular carcinoma (HCC) is the most common primary liver cancer worldwide, and early pathological diagnosis is crucial for formulating treatment plans. Despite the widespread attention to pathology in the treatment of HCC patients, a large amount of information contained in pathological images is often overlooked.

**Methods:** We retrospectively collected clinical data and pathological slide images from (a) 331 HCC patients at Qingdao University Affiliated Hospital between January 2013 and December 2016 and (b) 180 HCC patients from The Cancer Genome Atlas (TCGA). After data screening, precise quantification of various cell types was achieved using QuPath software. Key factors related to the survival prognosis of pathologically confirmed HCC patients were identified through Cox regression and neural network models, and potential therapeutic targets were screened.

**Results:** Our study showed that tumour-infiltrating lymphocytes (TILs) had a protective effect. We quantified the TILs index by machine learning and built a neural network model to predict the prognostic risk of patients (ROC = 0.836 for training set ROC validation set). 95% CI [0.7688–0.896], and there was a significant difference in prognosis in the high-low risk group predicted by the model ($p = 2.6\mathrm{e}{-18}$, HR = 0.18, 95% CI [0.12–0.27], and TNFSF4 was identified as a possible immunotherapy target.

**Conclusion:** This study included a total of 511 patients, divided into a training cohort of 331 cases (from Qingdao University Hospital between January 2013 and December 2016) and a validation cohort of 180 cases (TCGA). The results revealed that tumor-infiltrating lymphocytes (TILs) have a protective effect and successfully predicted the survival risk of liver cancer patients using machine learning and neural network technology. The discovery of TNFSF4 provides a new potential target for immunotherapy.

# INTRODUCTION

Hepatocellular carcinoma (HCC) is the most common malignant tumor of the digestive system (*Singal, Kanwal & Llovet, 2023*). It was estimated that 905,700 people worldwide were diagnosed with liver cancer, and 830,200 died from it (*Galle et al., 2019*).

In modern medical research, the prognosis of liver cancer patients after surgery varies significantly (*Kuo et al., 2022*). Some patients have better survival rates, while others face more challenges. Liver cancer surgery provides a unique opportunity to treat this disease.

In previous studies, the information hidden in pathological slides was often overlooked, but in reality, these slides may contain critical biomarkers (*Xie et al., 2021*; *Jiang et al., 2020*). A thorough analysis of these pathological slides can help us gain more insights into the microstructure and biological characteristics of liver cancer (*Melo et al., 2019*), which is crucial for predicting patient survival and providing personalized treatment plans (*Huang et al., 2023*).

To systematically explore the information contained in these pathological slides, We employed digital scanning methods to convert the slides into digital formats (*Hanna et al., 2022*; *Lee et al., 2021*), providing a foundation for subsequent quantification and analysis. Traditionally, tumor staging, pathological grading, and molecular markers such as TP53 and AFP have been widely used in the prognosis assessment of HCC patients (*Yang et al., 2021*). Tumor-infiltrating lymphocytes (TILs) have also been extensively studied, with high levels of TILs reported as favorable prognostic biomarkers (*Paijens et al., 2021*; *Fanale et al., 2022*). We have addressed these shortcomings by implementing digital scanning and machine learning techniques for digitizing pathological slides (*Fanucci et al., 2023*). Using the open-source software QuPath, We realized the automatic and quantitative analysis of TILs, such as the number, area and nucleo-plasmic ratio (*Acs et al., 2019*; *Bankhead et al., 2017*). TILs play a crucial role in tumor immune response, and their quantity and distribution patterns may significantly impact patient survival prognosis (*El Bairi et al., 2021*). By quantifying multiple indicators of TILs, we aim to build more comprehensive and accurate survival prediction models. This research model focuses on pathological information after surgical resection and makes full use of digital technology, incorporating machine learning and neural network models to provide more reliable support for individualised treatment and clinical decision-making for liver cancer patients. To optimize patient treatment plans, it is necessary to develop prognostic factors that can accurately distinguish high-risk and low-risk patients.

# MATERIALS AND METHODS

## Clinical cohorts

We retrospectively evaluated independent cases from the following two cohorts: (a) patients who underwent liver cancer resection surgery at Qingdao University Affiliated

Hospital from 2013 to 2016, and (b) The Cancer Genome Atlas (TCGA). We presented pathological images from both cohorts in whole-slide image (WSI) format. The Qingdao University Affiliated Hospital study cohort included 331 hepatocellular carcinoma (HCC) patients, corresponding to 331 digital pathological slides (WSIs). The study has been approved by the Ethics Committee of The Affiliated Hospital of Qingdao University (approval number QYFY WZLL 28523). After excluding poor-quality images, we ultimately retained 285 high-quality digital pathological slides for subsequent in-depth analysis and research. We selected diagnostic slides for download from the TCGA cohort, which included 378 HCC patients (https://portal.gdc.cancer.gov/analysis_page?app=Downloads, accessed on November 27, 2022).

In the cohort from (a) Qingdao University Affiliated Hospital, patients' risk characteristics (including test results) were obtained from admission to surgery, and pathological report data came from professional pathology tests.

## Ethics statement

The studies involving human participants were reviewed and approved by The Affiliated Hospital of Qingdao University Research Ethics Committee approved the study protocol. The patients/participants provided their written informed consent to participate in this study.

## Digital scanning

For each patient's diagnostic slide from the Qingdao University Affiliated Hospital study cohort, we selected three diagnostic slides per patient from a nine-point sample based on site. To ensure the quality and representativeness of the slides, we leveraged the expertise of two experienced pathologists. During the selection process, our criteria were as follows:

1. The sections are clear and without blurriness.
2. The tissue structure is clear, with good staining of cell nuclei and cytoplasm.
3. Abundant and well-defined cancer nests with clear layers.

After the selection was completed, we utilised the NanoZoomer-XR C12000 Hamamatsu scanner at a 20× magnification (0.5 μm/pixel) to ensure the clarity and resolution of the images.

## Training cell classifier for cell annotation

In our study, we utilised the open-source software QuPath, which integrates machine learning algorithms, to conduct digital analysis on whole-slide images (WSI). The selection of regions of interest (ROIs) was based on the positional characteristics of the tumour and was performed by two independent pathologists to ensure consistency in selection. Ultimately, we took the intersection of the ROIs selected by both experts, ensuring that the chosen ROIs covered at least 50% of the tumour area while excluding the fibrous capsule surrounding the tumor. Before conducting the research, we performed image normalisation and intensity correction. Subsequently, we used QuPath for cell identification within the ROIs (parameters are detailed in the Supplemental Material), followed by detailed annotation of the tumour area. When classifying all cells within the

ROI, we categorised cells into four main groups: red for tumour cells, yellow for lymphocytes, blue for stromal cells, and black for other cells (Fig. 1B).

We trained an efficient cell classifier (with eight hidden layers (maximum number of iterations: 1,000)) by manually annotating cells, enabling it to automatically classify all cells within the ROI. To achieve more accurate cell classification, we conducted multiple rounds of training (Fig. 1B). Finally, after independent evaluation by two pathologists, the QuPath-trained cell classifier was considered sufficiently accurate for further analysis and research when it reached an accuracy rate of 95%. However, there were variations in staining effects and tumour differentiation levels across different cancer nests. In cases where there were significant differences in staining effects and tumour differentiation levels among different cancer nests, and if the classifier's accuracy did not meet expectations during training, model performance could be optimised by manually correcting some misclassified areas. This process can be considered as one cycle of model iteration.

## Quantification of TILs indices

We adopted four currently recognized indices to quantify the presence of tumor-infiltrating lymphocytes (TILs):

**eTILs% (Percentage of TILs in Tumor Cells):**

**Definition:** Calculate the proportion of TILs in tumor cells.
**Formula:** eTILs% = (TILs/(TILs + Tumor Cells)) × 100

**esTILs% (Percentage of TILs in Stromal Cells):**

**Definition:** Calculate the proportion of TILs in stromal cells.
**Formula:** esTILs% = (TILs/(TILs + Stromal Cells)) × 100

**etTILs% (Percentage of TILs in Total Cells):**

**Definition:** Calculate the proportion of TILs in total cell count.
**Formula:** etTILs% = (TILs/Total Cell Count) × 100

**eaTILs (Density of TILs in Tumor Area):**

**Definition:** Calculate the infiltration density of TILs in the tumor area.
**Formula:** eaTILs = TIL/Analyzed Tumor Area ($mm^2$)

These indices, calculated with precise methods, help us comprehensively quantify the presence of TILs and reveal their distribution and density in the tumor microenvironment (*Zheng et al., 2022*).

## Statistical analysis

We utilised the Neuralnet and NeuralNetTools packages in R, version 4.3.2 (*R Core Team, 2023*), to incorporate the quantified pathological slide features for constructing a neural network model. Through the model, we obtained the high- and low-risk groupings for the

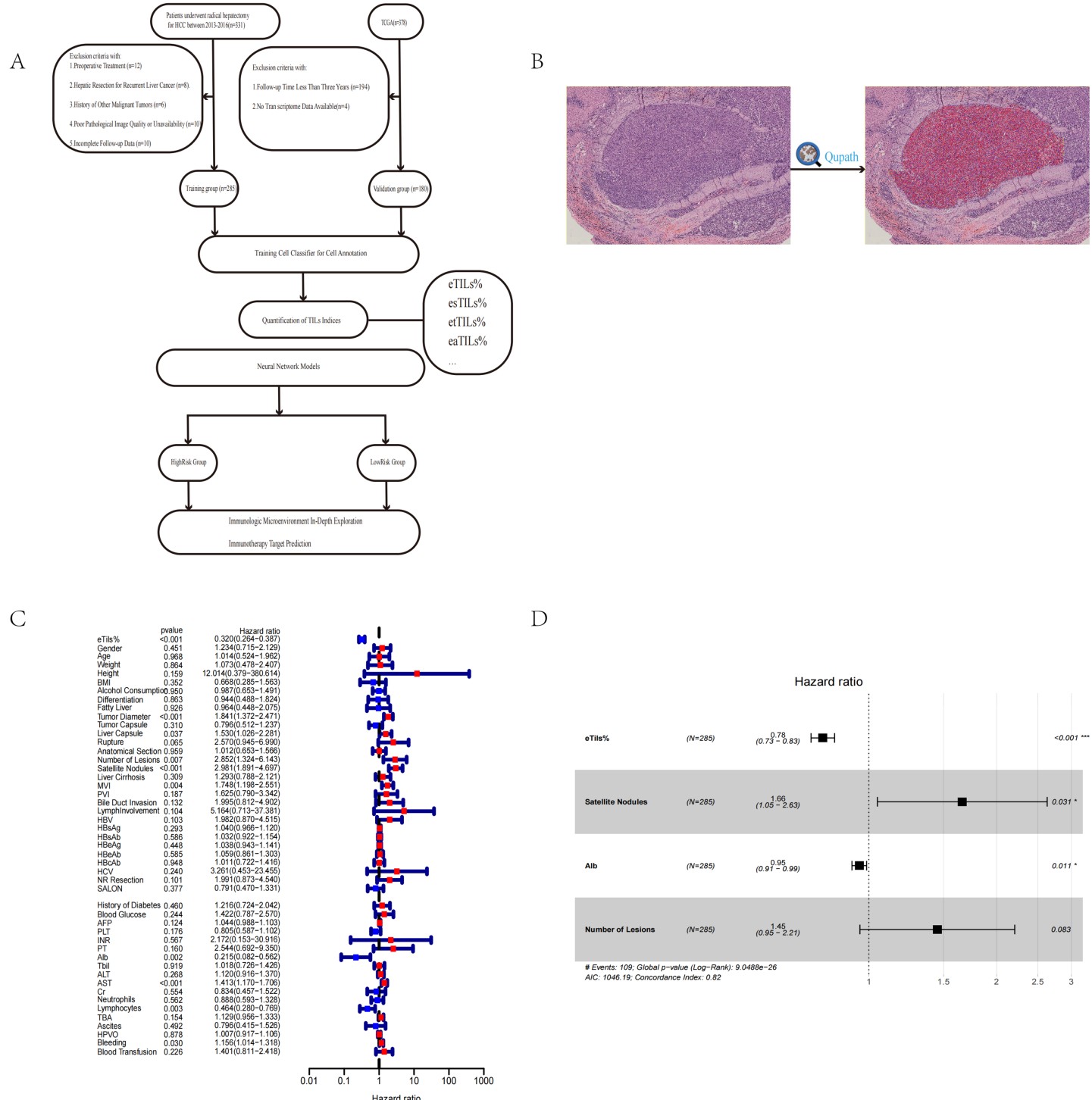

**Figure 1** **Workflow, software quantification interface, and Cox analysis.** (A) The flow chart of the study process. (B) Interface for quantifying TILs using QuPath. (C, D) Univariate and multivariate Cox analyses to evaluate independent prognostic factors of HCC. Tumor capsule, whether the tumor capsule is intact; liver capsule, whether the tumor invades the liver capsule; rupture, whether the tumor has ruptured; anatomical section, whether anatomical section was performed during surgery; MVI, microvascular invasion; PVI, portal vein tumor thrombus; LymphInvolvement, presence of lymphocytic infiltration around tumor nests; NR resection, whether it is radical resection; SALON, surgical method (laparoscopic or not); AFP, alpha-fetoprotein; PLT, platelets; INR, international normalized ratio; PT, prothrombin time; ALB, albumin; Tbil, total bilirubin; ALT, alanine aminotransferase; AST, aspartate aminotransferase; Cr, creatinine; HPVO, hepatic portal vein occlusion; TBA, total bile acids.

patients. Subsequently, we further utilised the IOBR and survival packages in R for subsequent statistical analyses, including immune infiltration, immune checkpoint prediction, immune scoring, as well as univariate and multivariate Cox regression analyses.

## Immunohistochemistry (IHC)

We performed IHC staining on 51 paraffin-embedded liver cancer tissues collected from Qingdao University Affiliated Hospital between January 2015 and June 2015 to determine the expression level of TNFSF4 protein. The staining intensity (0 = negative, 1 = weak, 2 = moderate, or 3 = strong) and the percentage of strongly stained cells in the observed field (0 = 0%, 1 = 1–25%, 2 = 26–50%, 3 ≥ 51%) were used to calculate the IHC staining score by multiplying the intensity and percentage scores. A final score <4 was defined as low expression, and a score of 4–9 was defined as high expression.

## RESULTS

### Patient characteristics

This study included a total of 465 HCC patients from two cohorts (median [IQR] age range of 57 [49–65] years for Cohort A and 61 [51–70] years for Cohort B; gender ratio ranged from 241 males [84.6%] out of 285 patients in Cohort A to 117 males [65%] out of 180 patients in Cohort B). The majority of diagnoses were for stages II and III of the disease. The datasets from both cohorts had similar distributions of clinical and pathological characteristics, and Table 1 summarises the baseline characteristics of our study cohorts (Table 1).

### Risk factor analysis

In the risk factor analysis, we first focused on the patient cohort from Qingdao University Affiliated Hospital. By tracking follow-up data until January 1, 2022, we obtained the survival time and survival status of the patients and conducted comprehensive statistics on numerous clinical data.

In this study, we used eTILs% (percentage of tumor-infiltrating lymphocytes in tumor cells) as a feature of TILs, and employed the Cox proportional hazards model to identify significant risk factors in survival analysis ($P < 0.05$) (Fig. 1C). These factors included eTILs%, satellite lesions (surgical record/pathology), tumor diameter, AST, microvascular invasion, lymphocytes, number of lesions, liver capsule, *etc.* (Table 2).

Through multivariate Cox regression analysis, we further revealed that eTILs%, satellite lesions, and ALB remained significant independent prognostic factors (Fig. 1D) (Table 3).

### The protective effect of TILs

In our study, through multivariate Cox analysis, we observed that the features of TILs exhibited the most significant differences in prognosis, with the highest hazard ratio (HR) and the smallest *p*-value. This highlights the importance of TILs in the survival analysis of liver cancer patients. To further explore the clinical significance of TILs and verify their independence in prognosis, we conducted subgroup analyses with two other features (ALB (Figs. 2A–2C) and satellite lesions (surgical record/pathology) (Figs. 2D, 2E)), confirming

**Table 1 Clinicopathologic characteristics of patients by cohort.**

| Characteristic | Cohort A (n = 285) (%) | Cohort B (n = 180) (%) |
|---|---|---|
| Sex | | |
| Female | 44 (15.4) | 63 (35) |
| Male | 241 (84.6) | 117 (65) |
| Age, median [IQR] | 57 [49–65] | 61 [51–70] |
| Cancer Stage | | |
| I | 164 (57.5) | 78 (43.3) |
| II | 99 (34.7) | 36 (20) |
| III | 22 (7.7) | 51 (28.3) |
| IV | 0 | 3 (1.7) |
| Unknown | 0 | 12 (6.7) |
| Tumor grade | | |
| 1 | 31 (10.9) | 25 (13.9) |
| 2 | 160 (56.1) | 81 (45) |
| 3 | 92 (32.2) | 64 (35.6) |
| 4 | 2 (0.7) | 7 (3.9) |
| Unknown | 0 | 3 (1.7) |

**Table 2 Univariate Cox analysis of queued patient information from Qingdao University Affiliated Hospital.**

| id | HR | HR.95L | HR.95H | p value |
|---|---|---|---|---|
| eTils% | 0.319510116 | 0.26410987 | 0.386531235 | 7.62E−32 |
| Gender | 1.233622648 | 0.714853295 | 2.128863151 | 0.450742369 |
| Age | 1.013676244 | 0.523688749 | 1.962118777 | 0.967844878 |
| Weight | 1.073098429 | 0.478402497 | 2.407053151 | 0.86409435 |
| Height | 12.01389767 | 0.379213219 | 380.6136755 | 0.158537132 |
| BMI | 0.667772047 | 0.285358566 | 1.562663817 | 0.351906246 |
| Alcohol consumption | 0.98696188 | 0.653496002 | 1.490588695 | 0.950253778 |
| Differentiation | 0.943607363 | 0.488037397 | 1.82443981 | 0.863003941 |
| Fatty liver | 0.964330842 | 0.448141648 | 2.075089374 | 0.925987203 |
| Tumor diameter | 1.841141959 | 1.371821401 | 2.471024078 | 4.79E−05 |
| Tumor capsule | 0.795821071 | 0.51212676 | 1.236668784 | 0.309885157 |
| Liver capsule | 1.529931449 | 1.026172092 | 2.28099191 | 0.036910777 |
| Rupture | 2.569876346 | 0.944762384 | 6.990397319 | 0.064505351 |
| Anatomical section | 1.011593146 | 0.653317692 | 1.566344682 | 0.958790947 |
| Number of lesions | 2.852487383 | 1.324474101 | 6.143332111 | 0.00740862 |
| Satellite nodules | 2.980615518 | 1.891425099 | 4.697023885 | 2.52E-06 |
| Liver cirrhosis | 1.293145077 | 0.788364735 | 2.121130126 | 0.308598431 |
| MVI | 1.748071617 | 1.19791758 | 2.550888668 | 0.00377364 |
| PVI | 1.625332466 | 0.790471558 | 3.341936339 | 0.186616197 |
| Bile duct invasion | 1.995264717 | 0.81206183 | 4.902436176 | 0.132047046 |

(Continued)

| id | HR | HR.95L | HR.95H | p value |
|---|---|---|---|---|
| **Table 2 (continued)** | | | | |
| LymphInvolvement | 5.163555123 | 0.713264329 | 37.38067419 | 0.104077265 |
| HBV | 1.981862926 | 0.869869858 | 4.515365858 | 0.103495651 |
| HBsAg | 1.040285961 | 0.966401427 | 1.119819209 | 0.293376042 |
| HBsAb | 1.031624845 | 0.922251054 | 1.153969753 | 0.586097754 |
| HBeAg | 1.037577407 | 0.943249229 | 1.141338728 | 0.448119464 |
| HBeAb | 1.059305462 | 0.861328306 | 1.302787861 | 0.58520967 |
| HBcAb | 1.011282609 | 0.722103984 | 1.416267654 | 0.947943923 |
| HCV | 3.260814992 | 0.453339774 | 23.45462503 | 0.240349481 |
| NR resection | 1.991111684 | 0.873240084 | 4.540018044 | 0.101494584 |
| SALON | 0.790983763 | 0.469983601 | 1.331227965 | 0.377343909 |
| History of diabetes | 1.215874605 | 0.724006013 | 2.041904387 | 0.459918815 |
| Blood glucose | 1.421996337 | 0.786763027 | 2.570117699 | 0.243693221 |
| AFP | 1.043871512 | 0.988249149 | 1.102624509 | 0.124326672 |
| PLT | 0.804551742 | 0.587306447 | 1.10215631 | 0.175657955 |
| INR | 2.172191567 | 0.152622683 | 30.91556319 | 0.566950436 |
| PT | 2.544385503 | 0.692424698 | 9.349605246 | 0.159596954 |
| Alb | 0.215057185 | 0.08228456 | 0.562068906 | 0.001716624 |
| Tbil | 1.017648564 | 0.726358368 | 1.425754347 | 0.919006651 |
| ALT | 1.120173541 | 0.916197146 | 1.369561963 | 0.268490376 |
| AST | 1.412947951 | 1.170144714 | 1.706132489 | 0.000326535 |
| Cr | 0.833764197 | 0.456721313 | 1.522072033 | 0.553829175 |
| Neutrophils | 0.887616736 | 0.593116713 | 1.328344746 | 0.562196958 |
| Lymphocytes | 0.463857336 | 0.279794524 | 0.769005857 | 0.002898393 |
| TBA | 1.1286698 | 0.955590736 | 1.333097391 | 0.154121609 |
| Ascites | 0.796044265 | 0.41532599 | 1.525756847 | 0.491973599 |
| HPVO | 1.007400524 | 0.917211573 | 1.10645771 | 0.877545561 |
| Bleeding | 1.156178712 | 1.013852899 | 1.318484384 | 0.030369739 |
| Blood transfusion | 1.400583592 | 0.811428507 | 2.417507375 | 0.226409511 |

**Note:**

Annotation: Tumor capsule, whether the tumor capsule is intact; liver capsule, whether the tumor invades the liver capsule; rupture, whether the tumor has ruptured; anatomical section, whether anatomical section was performed during surgery; MVI, microvascular invasion; PVI, portal vein tumor thrombus; LymphInvolvement, presence of lymphocytic infiltration around tumor nests; NR resection, whether it is radical resection; SALON, surgical method (laparoscopic or not); AFP, alpha-fetoprotein; PLT, platelets; INR, international normalized ratio; PT, prothrombin time; ALB, albumin; Tbil, total bilirubin; ALT, alanine aminotransferase; AST, aspartate aminotransferase; Cr, creatinine; HPVO, hepatic portal vein occlusion; TBA, total bile acids.

the significant prognostic value of eTILs% in different subgroups. This further substantiates that high levels of TILs infiltration can significantly extend patient survival time.

Subsequently, we performed Kaplan-Meier (K-M) curve analyses based on the four characteristics of TILs, all of which showed $p$-values less than 0.01. This indicates that, whether considering the overall level of TILs or their different characteristics, high levels of TILs infiltration are significantly associated with better patient survival outcomes.

**Table 3 Multifactorial Cox analysis of queued patient information from Qingdao University Affiliated Hospital.**

| ID | Univariate Cox analysis | | | | Multifactorial Cox analysis | | | | |
|---|---|---|---|---|---|---|---|---|---|
| | HR | HR.95L | HR.95H | $p$ value | coef | HR | HR.95L | HR.95H | $p$ value |
| eTils% | 0.319510116 | 0.26410987 | 0.386531235 | 7.62E−32 | −0.249536311 | 0.779161988 | 0.731038926 | 0.83045291 | 1.70E−14 |
| Satellite Nodules | 2.980615518 | 1.891425099 | 4.697023885 | 2.52E−06 | 0.506897695 | 1.66013296 | 1.047771695 | 2.630383562 | 0.03087371 |
| Alb | 0.215057185 | 0.08228456 | 0.562068906 | 0.001716624 | −0.051735057 | 0.949580418 | 0.912608765 | 0.988049868 | 0.010670927 |
| Number of lesions | 2.852487383 | 1.324474101 | 6.143332111 | 0.00740862 | 0.371409301 | 1.449776347 | 0.95273288 | 2.206128813 | 0.082933461 |

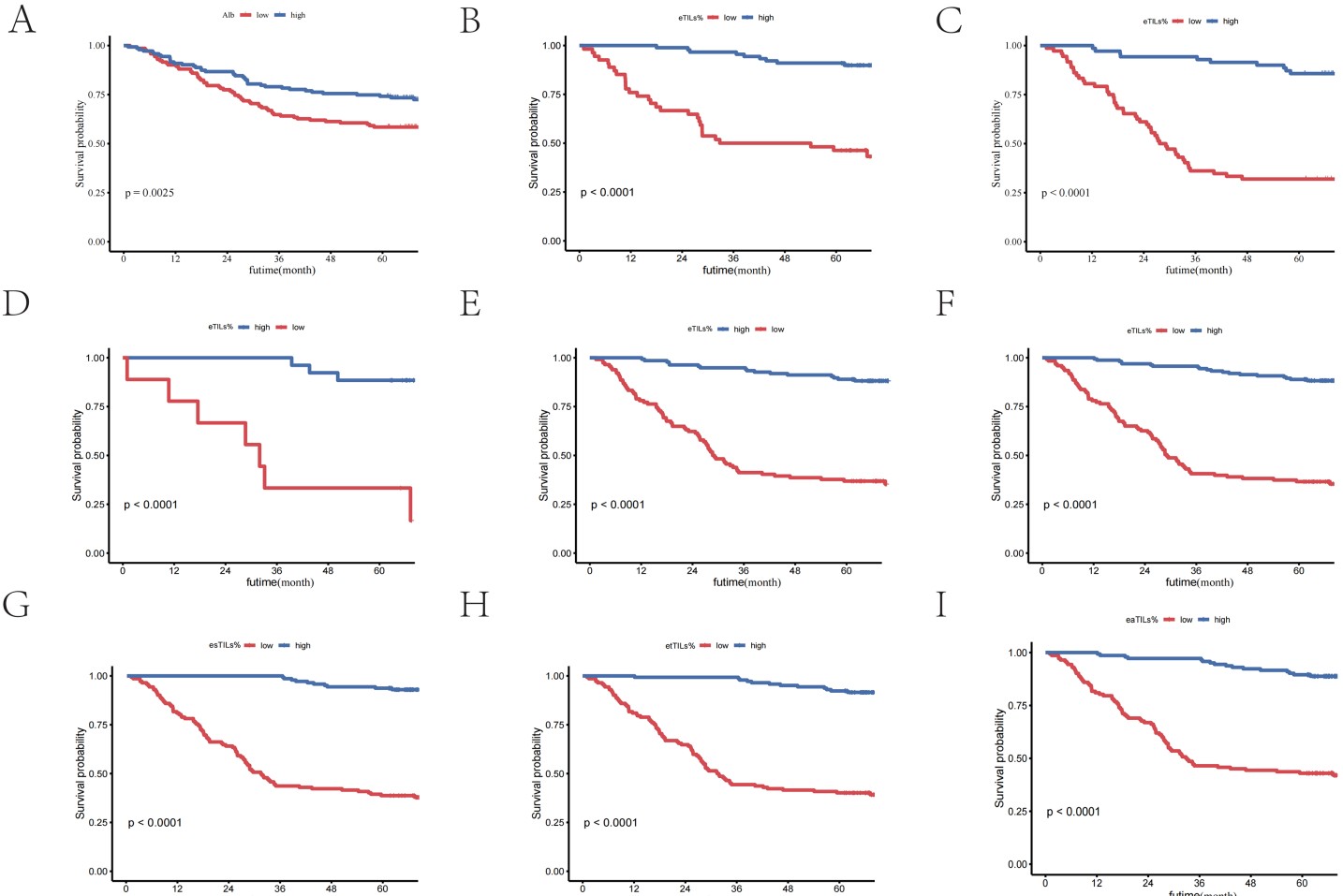

**Figure 2 TILs are independent prognostic factors in HCC patients.** (A) K-M curve of eTILs% stratified by median ALB. (B) K-M curve of eTILs% in the high ALB group. (C) K-M curve of eTILs% in the low ALB group. (D) K-M curve of eTILs% in patients with satellite nodules. (E) K-M curve of eTILs% in patients without satellite nodules. (F–I) K-M curves of the four features of TLSs.

## Neural network model

Based on the patient slide characteristics from the Qingdao University Affiliated Hospital study cohort, we utilized seven quantified features related to TILs, including four TILs

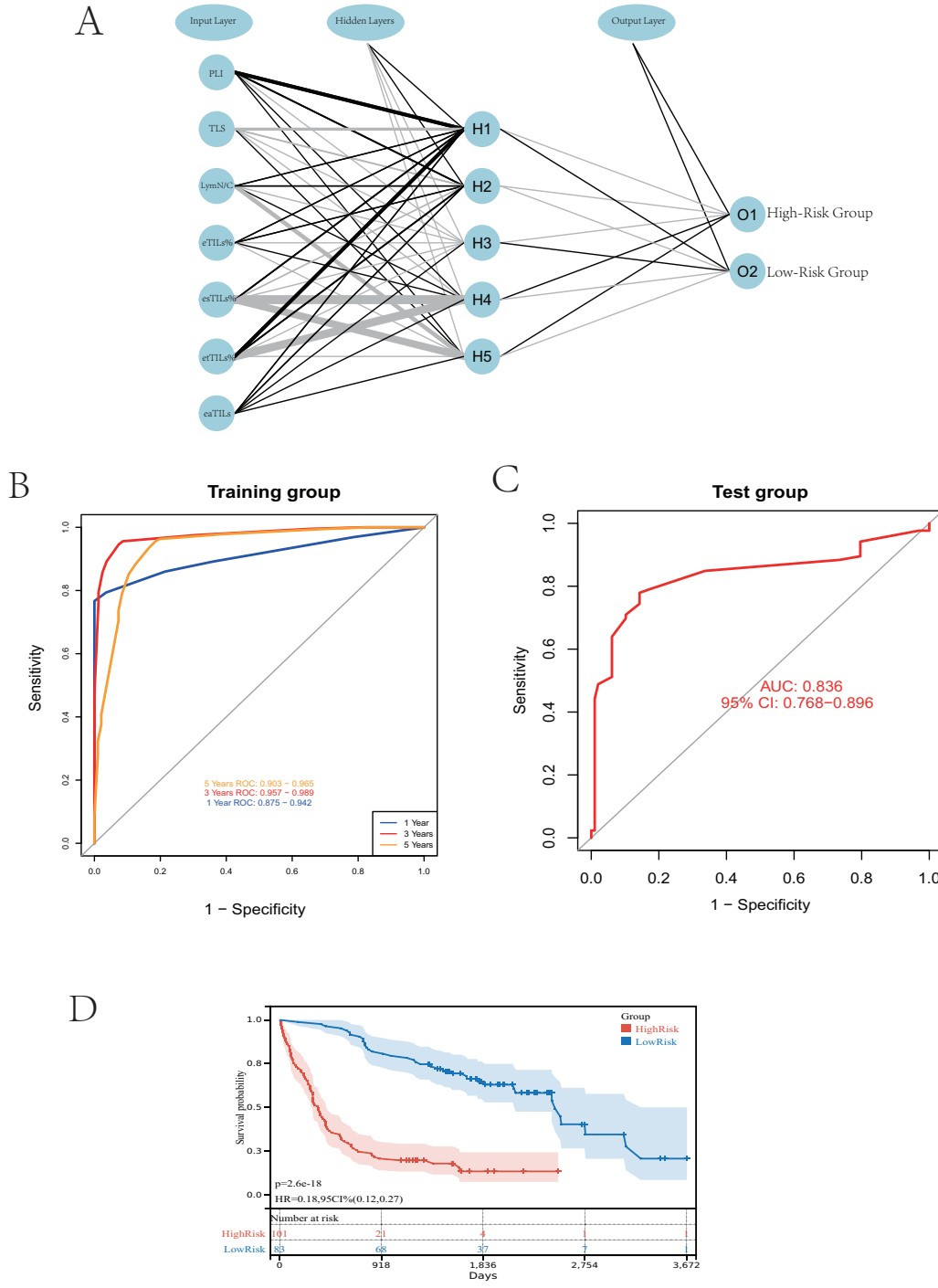

**Figure 3 Establishing a neural network model.** (A) Schematic diagram of the neural network model. (B) Performance comparison at different time cutoffs. (C) ROC curves of the validation group. (D) K-M curves of the high and low-risk groups.

features, the nuclear-cytoplasmic ratio of TILs, the presence of TILs-aggregated lymph nodes within cancer nests, and the infiltration of TILs within cancer nests. The model was constructed using the neuralnet package in R to build a multilayer perceptron (MLP)

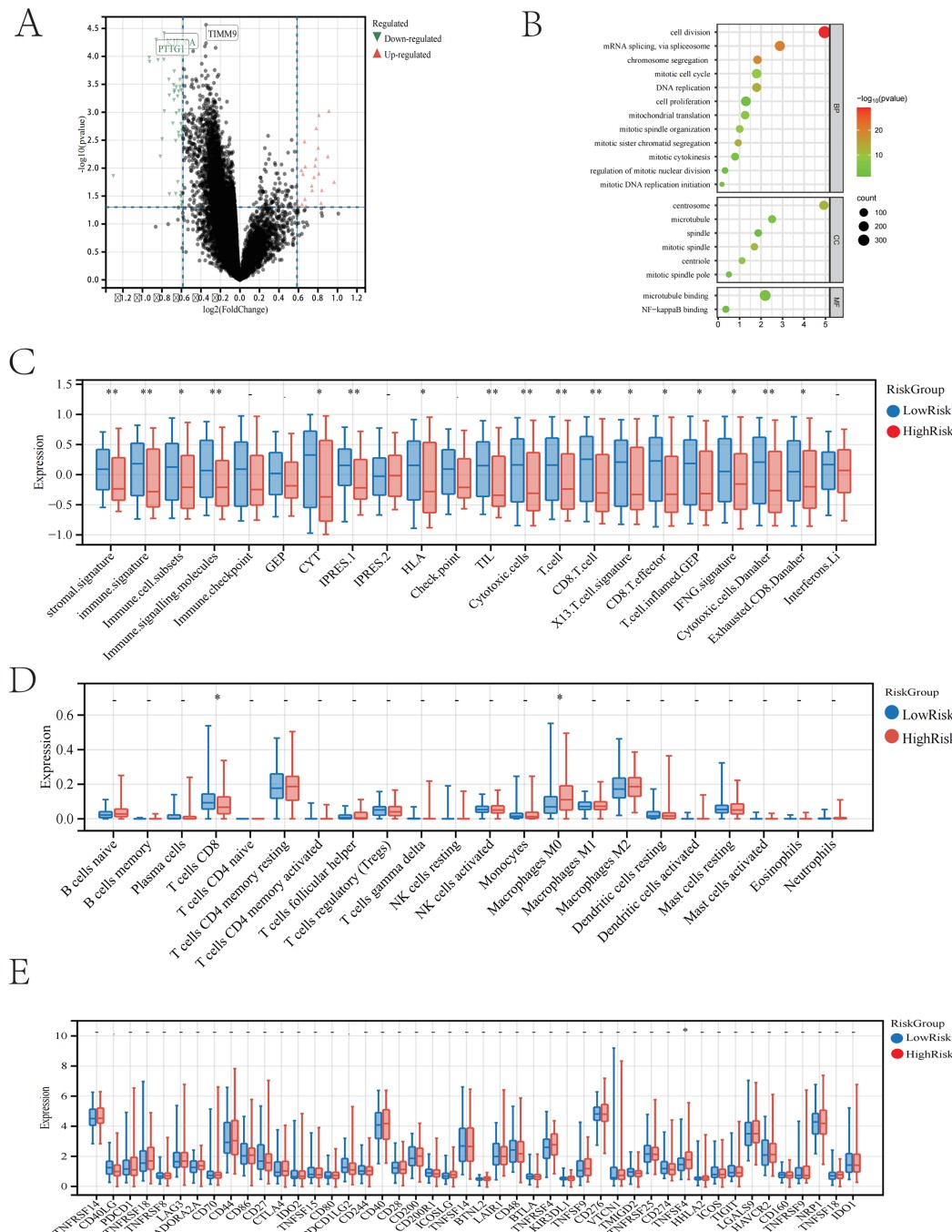

**Figure 4** **The neural network model predicts patients in the high and low-risk groups.** (A) Volcano plot of differentially expressed genes between the two groups. (B) GO enrichment analysis of differentially expressed genes between the two groups. (C) Immune-related scores in the high and low-risk groups. (D) Differences in immune cells between the high and low-risk groups. (E) Selection of immune checkpoints in the high and low-risk groups.

neural network, which includes one hidden layer with five neurones (hidden = 5), and the output layer corresponds to the risk stratification of patients (high risk and low risk) to predict their survival status (Fig. 3A).

**Table 4 The immunohistochemistry group's patients exhibited certain clinical baseline characteristics.**

| Characteristic | Low expression group ($n = 30$) | High expression group ($n = 20$) | $p$-value |
|---|---|---|---|
| Sex (Male %) | 85% | 85% | 0.75 |
| Age (mean ± SD) | 55.6 ± 11.4 | 60.2 ± 10.1 | 0.09 |
| Weight (kg) | 73.2 ± 10.9 | 64.7 ± 8.4 | <0.01 |
| Height (cm) | 168.3 ± 6.8 | 167.1 ± 6.3 | 0.45 |
| Alcohol consumption (%) | 30% | 30% | 1.00 |
| Differentiation | 2.5 ± 0.5 | 2.6 ± 0.6 | 0.52 |
| Fatty liver (%) | 10% | 5% | 0.50 |
| Tumor size (cm) | 3.6 ± 2.8 | 4.9 ± 2.7 | 0.06 |

The survival time endpoints were chosen at 1, 3, and 5 years according to mainstream standards (*Li & Ma, 2021*; *Albano, Bilfinger & Nemesure, 2018*). Following model validation, the ROC for the 3-year model reached 0.975, indicating the best performance. Therefore, we chose 3 years as the survival time endpoint for patients (Fig. 3B).

We then selected 378 patients from the TCGA training cohort, excluding those with follow-up times less than 3 years and without transcriptome data, resulting in 180 patients for the external validation set. We performed the same cell segmentation and annotation processes on the validation set, resulting in a final ROC value of 0.836, indicating the successful construction and performance of our model in external validation (Fig. 3C). This professional analysis emphasizes the model's reliability and applicability, providing strong support for future applications of TILs in liver cancer prognosis research.

After completing the neural network model validation, we classified the patients in the TCGA validation cohort into high-risk and low-risk categories based on the model results, and subsequently performed Kaplan-Meier (K-M) survival curve analysis. The K-M curve showed significant results, with a $p$-value less than 0.01, further proving the model's accuracy in risk stratification (Fig. 3D).

## Immune analysis and checkpoint screening

We conducted an in-depth study on the gene expression differences between two groups of patients predicted by the model in the TCGA cohort, filtering out 44 differentially expressed genes that are highly expressed in the high-risk group according to the criteria of $p < 0.05$ and LogFC $< -0.5$ (Fig. 4A). After obtaining the set of differential genes, we performed gene functional enrichment analysis using tools such as the clusterProfiler package in R, which indicated that these genes play a key role in biological processes involving cell proliferation and cell division (Fig. 4B). We speculate that these genes may promote further division and proliferation of tumour cells, leading to adverse outcomes. Subsequently, we conducted immune scoring for both high- and low-risk groups, and the analysis showed significant differences in the low-risk group in aspects such as stromal signature, immune signature, immune cell subsets, immune signalling molecules, immune CYT, IPRES.1, HLA, TIL, cytotoxic cells, T cells, CD8 T cells, X13 T cell signature, CD8 T
A

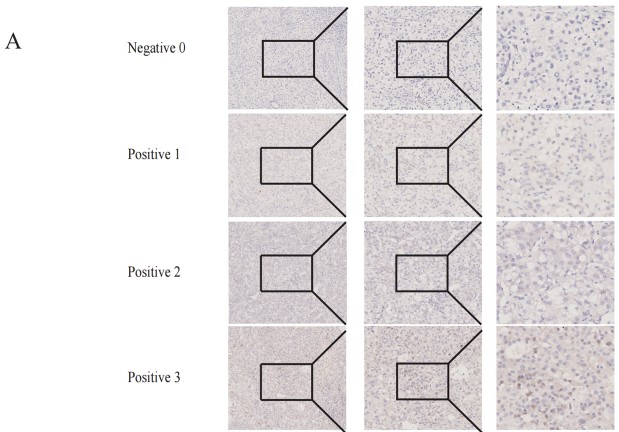

B

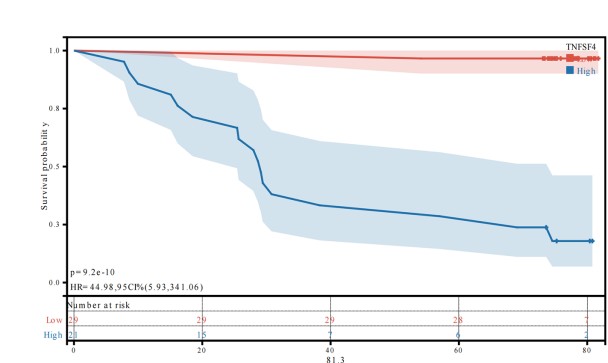

C

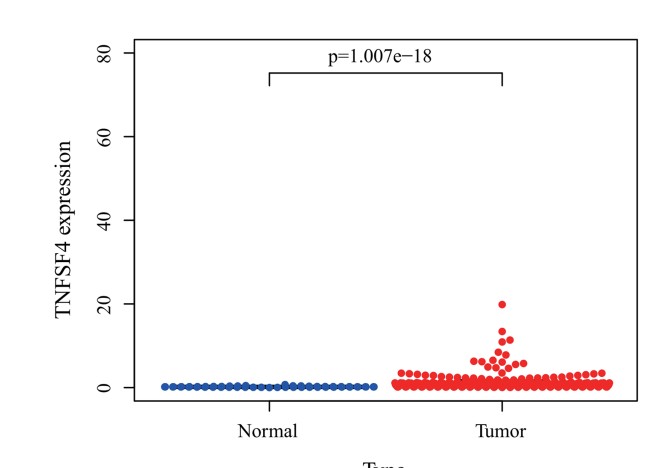

D

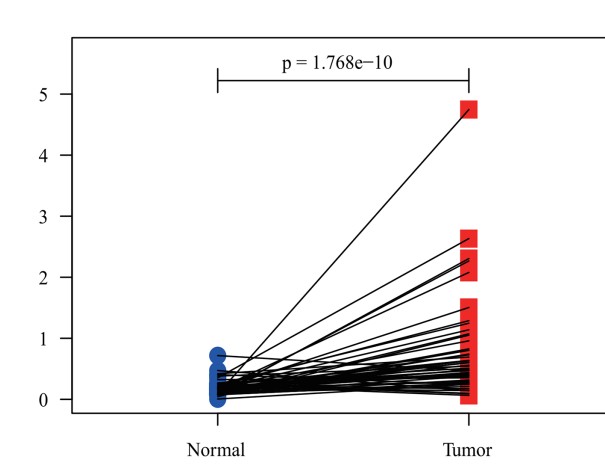

**Figure 5 Poor prognosis is linked to TNFSF4 high expression in HCC.** (A) Immunohistochemical analysis of TNFSF4 protein expression in human HCC tissue. (B) TNFSF4 protein expression in 116 patients with HCC was correlated with OS. (C) Differences in TNFSF4 mRNA expression between HCC tissues and normal tissues in patients with HCC in the TCGA database. (D) TNFSF4 mRNA expression differences between hepatocellular carcinoma tissue and adjacent normal tissue in paired samples from the TCGA database.

effector, nd inflamed T cells. GEP, IFNG signature, cytotoxic cells. Danaher, and exhausted. CD8.Danaher (Fig. 4C). Immune infiltration analysis revealed a higher presence of macrophages M2 in the low-risk group, suggesting that the immune response may be associated with the polarisation of macrophages M2 (Fig. 4D) (*Cao et al., 2023*).

Finally, we conducted an immune checkpoint analysis based on the differences in TILs using the aforementioned differential genes, screening for 46 immune checkpoints. Among these checkpoints, we identified TNFSF4 as a potential target that may benefit, offering possible assistance for immune therapy in liver cancer patients (Fig. 4E).

## TNFSF4 in IHC and expression analysis

To further validate our bioinformatics findings that identified TNFSF4 as a potential therapeutic target, we selected pathological slides from 50 consecutive patients within 1 year from the Qingdao University Affiliated Hospital study cohort (Table 4). We re-sectioned these patients' paraffin blocks and performed immunohistochemistry (IHC) staining using a TNFSF4 antibody (Fig. 5A).

Through IHC staining, we found that the expression levels of TNFSF4 in HCC patients were significantly associated with prognosis. There was a significant difference in prognosis between the high expression and low expression groups ($p < 0.05$) (Fig. 5B). This result suggests that TNFSF4 could serve as a potential biomarker for the prognosis of HCC patients.

To further confirm the expression pattern of TNFSF4 in HCC, we analyzed data from The Cancer Genome Atlas (TCGA) database. The mRNA expression of TNFSF4 was much higher in HCC tissues compared to liver tissues that did not have a tumor (Fig. 5C). Paired sample analysis further validated this finding, revealing significantly higher TNFSF4 expression in tumor tissues compared to adjacent non-tumor tissues (Fig. 5D).

These findings indicate that TNFSF4 is not only highly expressed in HCC tissues, but that its high expression is also associated with a poor patient prognosis.

## Conclusion

This study looked back at liver cancer patients from two large research groups, using machine learning and neural network models to look at clinical data and digitally scanned pathological slides. We deeply investigated the relationship between the immune status of liver cancer patients and their prognosis. The study quantified TILs and other key factors in a precise manner, significantly improving the analysis's efficiency and accuracy. Our successfully established neural network model not only reliably predicted patient prognosis, but also performed excellently in independent validation cohorts, achieving high ROC results and demonstrating the model's robustness and generalizability. The study comprehensively examined the distribution of TILs across different cell types, covering various dimensions such as cell proportions and infiltration densities. In addition, the study looked closely at immune scores, immune infiltration, and immune checkpoints to find out how TILs are connected to the immune microenvironment in liver cancer and what that means for patients. The study used differential gene screening to find that TNFSF4 was highly expressed in the high-risk group. This suggests that TNFSF4 could be a personalized therapeutic target for immunotherapy and points the way for new treatment strategies in the future.

## DISCUSSION

Liver cancer is a complex and highly heterogeneous disease (*Llovet et al., 2021*). Studies have shown that the main factors affecting the survival of liver cancer patients are diverse (*Forner, Reig & Bruix, 2018*). Therefore, identifying high-risk patients and extending their survival time has become an important task. The pathological sections provided during

surgery for operable patients contain a wealth of valuable information due to the development of histopathology.

Tumor-Infiltrating Lymphocytes (TILs) refer to lymphocytes present in tumor tissues. TILs are an important component of the tumor immune microenvironment, capable of infiltrating the tumor and performing immune surveillance and attacks (*Schumacher & Schreiber, 2015*). TILs mainly include CD8+ cytotoxic T cells, CD4+ helper T cells, B cells, natural killer (NK) cells, and other types of immune cells (*Su et al., 2024*). TILs play a key role in the anti-tumor immune response. High levels of TIL infiltration are usually associated with a good prognosis because they can directly kill tumor cells or activate other immune cells by secreting cytokines (*Hendry et al., 2017*; *Bruni, Angell & Galon, 2020*). Therefore, the presence and activity of TILs can serve as biomarkers for cancer prognosis and treatment response (*Denkert et al., 2018*). Studies have found that high levels of TILs usually predict a better response to immunotherapy, such as PD-1/PD-L1 inhibitors (*Galon & Bruni, 2019*). In recent years, with the in-depth research on the tumor immune microenvironment and immunotherapy mechanisms, scientists are committed to enhancing the anti-tumor activity of TILs through strategies such as gene editing, co-stimulatory molecule modification, and immune checkpoint blockade, in order to improve the cure rate and survival rate of cancer patients (*June et al., 2018*; *Ribas & Wolchok, 2018*). Numerous studies have shown that the tumor microenvironment plays a crucial role in the growth and development of tumors. As an important component of the tumor microenvironment, TILs have attracted widespread attention.

In the pathological image analysis of liver cancer, the application of image analysis technology is receiving increasing attention. Unlike previous studies that focused more on the entire image (*Shi et al., 2021*), our research focuses on the quantification of lymphocytes in pathological images. To identify risk factors for liver cancer patients, we collected a large amount of clinical data and conducted a risk factor analysis based on numerous clinical features. Using large public datasets and our own patient cohort, we quantified the infiltration level of lymphocytes in pathological images as an important feature of the analysis. The neural network model we constructed can effectively predict the survival time of hepatocellular carcinoma (HCC) patients, achieving high-precision prediction of patient survival risk. At the same time, this model revealed the importance of TNFSF4 as a potential therapeutic target. This immune target identification provides new insights for HCC prognosis assessment and personalized treatment. Clinically, the level of TIL expression in the pathological images of HCC patients is an independent predictor of overall survival (OS).

Our study found that TILs play an important independent role in the prognosis of HCC patients. We confirmed using multivariate Cox regression analysis and Kaplan-Meier survival curve analysis that four TIL quantification indicators (eTILs%, esTILs%, etTILs%, and eaTILs%) were significantly linked to patient survival. eTILs% was the most significant predictor of survival across all subgroups. This indicates that TILs not only play an important role in the overall prognosis of liver cancer patients, but are also key prognostic factors in different clinical subgroups.
Neural network models are a type of computational model that is inspired by biological neural systems (*Kriegeskorte & Golan, 2019*). Research by *Altaf et al. (2024)* shows that one can predict the risk of liver cancer recurrence after liver transplantation using information such as patient clinical data through artificial intelligence technology (*Altaf et al., 2024*; *Nam et al., 2020*). They consist of a large number of interconnected artificial neurons, mimicking the connections and information transmission processes between neurons in the brain. Neural networks typically consist of three main parts: the input layer, the hidden layers, and the output layer. Most deep learning models or convolutional neural networks typically analyze whole slide images by dividing them into patches, which may introduce additional confounding factors. In contrast, our proposed neural network model focuses more on the quantification of tumor-infiltrating lymphocytes (TILs), directly quantifying their numbers to better illustrate their protective role in patients. This approach minimizes confounding factors and allows for a more accurate assessment of the relationship between TILs and patient prognosis. The TIL-based neural network model was very good at making predictions. It correctly predicted how long HCC patients would live and was a useful tool for figuring out their risk and giving them the right treatment. TNFSF4 (also known as OX40L) is an important immune checkpoint molecule that plays a key role in tumor immune escape (*Li et al., 2021a*). High expression of TNFSF4 may promote tumor cells to evade immune surveillance, leading to poor prognosis. Therefore, TNFSF4 is not only a potential prognostic marker but may also become a target for HCC immunotherapy. Future research should further explore the specific mechanisms of TNFSF4 in HCC and verify its potential as a therapeutic target.

In conclusion, this study shows that TILs are important as prognostic factors and suggests TNFSF4 as a new possible therapeutic target by looking at the numbers of TILs in pathological images of liver cancer patients. These findings provide new directions and methods for the individualized treatment of liver cancer.

Previous studies on TNFSF4 and TNFRSF4 have primarily focused on autoimmune diseases (*Moreno-Eutimio et al., 2021*; *Duffus et al., 2019*). Some studies have indicated that the expression of TNFSF4 can lead to drug resistance and tumor progression (*Roszik et al., 2019*; *Li et al., 2021b*). However, our study found that TNFSF4 might be a relevant immune checkpoint screened based on differences in TILs. Validation based on gene expression results may have limitations; future experimental and larger-scale clinical validations are necessary.

Despite achieving a series of significant results, this study has some limitations and shortcomings that warrant further discussion. First and foremost, as demonstrated by *Ding et al. (2021)*, AFP is a well-established predictor of prognosis in patients with hepatocellular carcinoma. While when performing regression analysis on the included risk factors, it was found that the traditional indicator for predicting HCC recurrence, AFP, was not significant in the univariate analysis. This may be related to the sample size. Additionally, there are studies pointing out that the postoperative decrease in alpha-fetoprotein (AFP), known as the AFP response (AR), can predict the efficacy of liver cancer surgery (*Zhou et al., 2019*). In this study, the dynamic changes of AFP were not considered, and only the static assessment of AFP levels was conducted, which may affect

its significance. In addition, this study relied on case cohorts from Qingdao University Affiliated Hospital and patient data from the TCGA database. Although we strived to ensure the quality of the research cohort, the sample size was relatively small, potentially limiting the model's generalizability. Future research should consider multi-center collaborations to expand the sample size. Second, the TCGA database contains diverse data from different laboratories and platforms. This diversity may lead to data heterogeneity in digital pathology image analysis and gene expression, affecting analysis consistency (*Howard et al., 2021*). Future work must consider and control this heterogeneity more effectively. Lastly, this study chose 3 years as the follow-up endpoint for patients based on the model's performance in the validation set. However, the choice of this time point may influence the results and requires further verification.

## ACKNOWLEDGEMENTS

The authors would like to express their deepest gratitude to Dr. Han and Professor Jin. Thanks to the teachers of the Affiliated Hospital of Qingdao University and the partners of Shanghai Jiao Tong University for their help.

### Funding

This work was supported by grants from the Science Technology Program of Shinan District of Qingdao City (grant number 2022-4-006-YY) and CSCO- MSD Oncology Research Fund Project (2022) (Y-MSDPU2022-0315). The funders had no role in study design, data collection and analysis, decision to publish, or preparation of the manuscript.

### Grant Disclosures

The following grant information was disclosed by the authors:
Science Technology Program of Shinan District of Qingdao City: 2022-4-006-YY.
CSCO- MSD Oncology Research Fund Project (2022): Y-MSDPU2022-0315.

### Competing Interests

The authors declare that they have no competing interests.

### Author Contributions

- Wenqing Zhong conceived and designed the experiments, performed the experiments, analyzed the data, prepared figures and/or tables, authored or reviewed drafts of the article, and approved the final draft.
- Ziyin Zhao conceived and designed the experiments, prepared figures and/or tables, authored or reviewed drafts of the article, and approved the final draft.
- Xin Fang performed the experiments, authored or reviewed drafts of the article, and approved the final draft.
- Jingyi Sun analyzed the data, prepared figures and/or tables, authored or reviewed drafts of the article, and approved the final draft.

- Yanbing Wei analyzed the data, prepared figures and/or tables, authored or reviewed drafts of the article, and approved the final draft.
- Fengda Li performed the experiments, authored or reviewed drafts of the article, and approved the final draft.
- Bing Han conceived and designed the experiments, authored or reviewed drafts of the article, and approved the final draft.
- Cheng Jin conceived and designed the experiments, authored or reviewed drafts of the article, and approved the final draft.

## Human Ethics

The following information was supplied relating to ethical approvals (*i.e.*, approving body and any reference numbers):

The studies involving human participants were reviewed and approved by The Affiliated Hospital of Qingdao University Research Ethics Committee approved the study protocol. The patients/participants provided their written informed consent to participate in this study.

## Data Availability

The raw measurements are available in the Supplemental Files.

## Supplemental Information

Supplemental information for this article can be found online at http://dx.doi.org/10.7717/peerj.19351#supplemental-information.

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
