# Peer review of "Constructing a neural network model based on tumor-infiltrating lymphocytes (TILs) to predict the survival of hepatocellular carcinoma patients"

_PeerJ, doi:10.7717/peerj.19351_

## Round 0.1 · original submission · Major Revisions

A major revision is needed to address the details put forward by the reviewers for further deliberation.

Reviewer 1 ·

Basic reporting

It is clear but the methodology needs to be more detailed.

Experimental design

The methodology needs more description.

Validity of the findings

Most data have been provided. The authors should include a table showing the study cohort's baseline characteristics.

Additional comments

Hepatocellular carcinoma (HCC) is the most common malignant tumor of the digestive system.
Please cite some credible reference
The study looked at pathological variables associated with outcomes. The main limitation is that there is very little treatment-altering value, as surgery has already been performed. However, the risk of recurrence might be calculated based on TILS, and this can guide subsequent treatment in the form of adjuvant immunotherapy, etc.
Sometimes, in one patient, there is a well differentiated tumor and a second moderate or poorly differentiated tumor. what was the selection criteria for slides in such patients.
AFP is one of the strongest indicators of post-transplant recurrence but was not significant in this study on univariate analysis. The authors should discuss why. Are any AFP cutoffs used for resection? (Figure 1 C). Why tumor diameter is not included in multivariate analysis even though it was significant in univariate (Figure D).
The legend and labeling in Figure 2 needs more details. The prognostic factor for what? The authors are looking at post-resection recurrence risk or survival.
The Kaplan Meier curve should be truncated at 5 years and the labeling should be in months or years to make it easy to understand.
There should a Table in results showing baseline patient characteristics such as age, sex, CTP score, tumor size, tumor number, etiology etc.
The authors should discuss and compare their outcomes with other neural network studies for HCC in the setting of resection or LT. They used SHAP values while this was not done in other studies. Any specific reasons?

Reviewer 2 ·

Basic reporting

The manuscript has potential but requires significant improvements to meet publication standards. The writing would benefit from clearer and more detailed descriptions of the methodologies, including the types of analyses performed and quantitative criteria applied for each type of analysis. Expanding the background section to engage more thoroughly with relevant literature would strengthen the contextual foundation of the study. Additionally, sharing raw data and code will improve reproducibility and transparency. A major revision is necessary to address these issues and strengthen the manuscript for consideration.

1. Abstract - Methods: The manuscript states 180 patients from the TCGA dataset were ultimately used for validation which is a relatively smaller number compared to the original 378. Reporting the number after filtering in this case ensures clarity and transparency about the basis of the results.
2. Introduction - Line 30: Please specify the year or duration these statistics were collected to provide temporal context for the data.
3. Minor - Line 45: The word "We" is capitalized in the middle of a sentence. Please correct this and ensure consistency in capitalization throughout the manuscript.
4. Minor - Line 95: The word "Formula" is misspelled. Please ensure spelling consistency throughout the manuscript.
5. General: Will the data and code be made available to enhance reproducibility?

Experimental design

The study identifies an important knowledge gap and emphasizes the potential of pathological slides to reveal critical biomarkers for hepatocellular carcinoma. However, the methods are not described with sufficient detail and clarity to enable replication, which is crucial for ensuring the robustness and reproducibility of the study. Strengthening the methodological descriptions would enhance the overall quality and impact of the manuscript.

6. Materials and Methods - Line 59: Selecting only high-quality slides may introduce selection bias, limiting generalizability. Please provide quantitative criteria or thresholds for excluding poor-quality images.
7. Materials and Methods - Line 64: Analyze excluded slides and perform sensitivity analysis to assess the impact of excluding lower-quality cancer nests on model performance and generalizability.
8. Materials and Methods - Line 67: Please provide staining intensity thresholds to ensure clarity and reproducibility.
9. Materials and Methods - Lines 74-77: The process of selecting regions of interest (ROIs) is subjective and lacks defined criteria. Explicitly stating the selection methodology (e.g., percentage of tumor area covered, staining thresholds) would improve reproducibility.
10. Materials and Methods - Line 78: What modules of QuPath software were used for cell identification? More details are required to ensure reproducibility.
11. Materials and Methods - Line 79: While staining variability is acknowledged, no specific measures to standardize or normalize staining effects (e.g., image preprocessing techniques) are mentioned.
12. Materials and Methods - Line 81: Please describe methods used to evaluate the performance of the cell classifier.
13. Materials and Methods - Lines 87-90: The authors state adjustments may be necessary for different cancer nests but do not describe how these adjustments are systematically applied. A standard protocol or automated adjustment strategy could improve reliability.
14. Statistical Analysis - Lines 107-110: Please provide more details on the neural network architecture. What were the inputs and outputs? What packages were used for subsequent statistical analyses such as immune infiltration and Cox analyses? A workflow diagram could enhance clarity.

Validity of the findings

15. Independent Prognostic Role of TILs - Line 131: Did the authors mean eTILs% have a protective effect implied by a low HR value? Please rephrase for clarity.
16. Neural Network Model - Lines 141-145: The neural network structure and training process need more technical details, such as the architecture, hyperparameters, and type of neural network model used to predict patient survival.
17. Neural Network Model - Lines 146-148: What is the rationale behind using the 3-year endpoint for survival prediction? Is there a biological or clinical reasoning for selecting this survival time endpoint?
18. Neural Network Model - Line 156: For the "high-risk and low-risk categories," please provide quantitative criteria for stratifying patients into these categories.
19. Immune Analysis and Checkpoint Screening - Lines 160-161: Was the gene expression analysis performed on the 51 paraffin-embedded liver cancer tissues? What were the number of genes differentially expressed? Was multiple testing correction applied? How was gene function enrichment analysis performed?

Additional comments

As there are multiple analysis performed in this study, consider providing a complete workflow diagram to clarify the sequence and relationship of analyses in the study.

---

## Round 0.2 · Minor Revisions

The authors need to further revise the article in accordance with the comments of Reviewer 1 before it is accepted.

Reviewer 1 ·

Basic reporting

Unambiguous, professional English is used throughout.

More references should be added.

Experimental design

The research question is is well defined, relevant, and meaningful. It states how the research fills an identified knowledge gap.

Validity of the findings

No comment.

Additional comments

The conclusion in the abstract can be shortened and the first two lines can be placed in the methods or results section. The result section should have actual numbers and percentages to show how the authors reached their conclusions. In the present form, the abstract does not contain any results.
This is a significantly large dataset of HCC patients, and the assumption that AFP was not significant due to the small number of HCC patients in this study should be explored further. What sort of measurement units were used, at what time AFP was performed before resection, etc.
1) Ding HF, Zhang XF, Bagante F, Ratti F, Marques HP, Soubrane O, Lam V, Poultsides GA, Popescu I, Alexandrescu S, Martel G, Workneh A, Guglielmi A, Hugh T, Aldrighetti L, Lv Y, Pawlik TM. Prediction of tumor recurrence by α-fetoprotein model after curative resection for hepatocellular carcinoma. Eur J Surg Oncol. 2021 Mar;47(3 Pt B):660-666. doi: 10.1016/j.ejso.2020.10.017. Epub 2020 Oct 15. PMID: 33082065.
2) Santambrogio R, Opocher E, Costa M, Barabino M, Zuin M, Bertolini E, De Filippi F, Bruno S. Hepatic resection for "BCLC stage A" hepatocellular carcinoma. The prognostic role of alpha-fetoprotein. Ann Surg Oncol. 2012 Feb;19(2):426-34. doi: 10.1245/s10434-011-1845-6. Epub 2011 Jul 6. PMID: 21732145.
The authors have added substantially to the discussion and limitations but mainly their comments and not supported by references to other studies. The authors might want to compare their findings with other studies dealing with HCC recurrence using neural networks but different methodologies and therefore identifying different prognostic factors. This would add value to the current study as to what it adds to the existing literature.
1) Altaf A, Mustafa A, Dar A, Nazer R, Riyaz S, Rana A, Bhatti ABH. Artificial intelligence-based model for the recurrence of hepatocellular carcinoma after liver transplantation. Surgery. 2024 Nov;176(5):1500-1506. doi: 10.1016/j.surg.2024.07.039. Epub 2024 Aug 23. PMID: 39181726.
2) Nam JY, Lee JH, Bae J, Chang Y, Cho Y, Sinn DH, Kim BH, Kim SH, Yi NJ, Lee KW, Kim JM, Park JW, Kim YJ, Yoon JH, Joh JW, Suh KS. Novel Model to Predict HCC Recurrence after Liver Transplantation Obtained Using Deep Learning: A Multicenter Study. Cancers (Basel). 2020 Sep 29;12(10):2791. doi: 10.3390/cancers12102791. PMID: 33003306; PMCID: PMC7650768.

Reviewer 2 ·

Basic reporting

No comment

Experimental design

No comment

Validity of the findings

No comment

Additional comments

Thank you for sharing the revised manuscript. I have reviewed the authors' responses to the comments and the corresponding changes made to the manuscript. I am pleased to note that the authors have addressed the points raised comprehensively.

I do not have any further comments at this time. Wishing you the best of luck with the next steps in the process!

---

## Round 0.3 · accepted · Accept

After revisions, one reviewer agreed to publish the manuscript. There was one reviewer left with a minor revision, and I think the author has responded adequately. I also reviewed the manuscript and found no obvious risks to publication. Therefore, I also approved the publication of this manuscript.